# Temperature-Activated Change of Permeable Material Properties for Low-Noise Trailing Edge Applications

Jonathan Mayer *, Alejandro Rubio Carpio * and Daniele Ragni *

AWEP Department, Delft University of Technology, 2629 HS Delft, The Netherlands
* Correspondence: j-h-mayer@web.de (J.M.); a.rubiocarpio@tudelft.nl (A.R.C.); d.ragni@tudelft.nl (D.R.)

**Abstract:** The present work analyses broadband noise scattering from permeable trailing edges with identical micro-structure but under a change of temperature. Experiments are performed in an anechoic wind tunnel using a NACA0018 airfoil at chord-based Reynolds numbers between $1.88 \times 10^5$ and $3.14 \times 10^5$ and no incidence. A microphone array is used to determine far-field sound pressure level changes upon trailing edge heating. It is found that broadband noise emission can be actively controlled by varying the temperature of the porous trailing edge inserts. Specifically, the electrically heated inserts yield a noise level variation of up to 2.5 dB with the heated one being noisier compared to a baseline, unheated material with similar micro-structure. Resistivity measurements of permeable samples with varying temperature show that flow resistivity increases with the fluid temperature which is in agreement with the reported trailing edge noise increase.

**Keywords:** porous materials; turbulent boundary layer trailing edge noise; microphone array; flow resistivity

## 1. Introduction

Turbulent boundary layer trailing edge (TBL-TE) noise can be a major contributor to airfoil self-noise in aerospace [1–3] and wind energy [4,5] applications. Far-field noise is emitted by scattering of convected turbulent boundary layer pressure fluctuations due to the jump in impedance at the trailing edge [6]. A promising concept for TBL-TE noise reduction, firstly investigated in the 70's [7,8], is the use of "variable impedance" airfoils for which the solid trailing edge is replaced with permeable materials. Later studies confirmed noise reduction capabilities of porous treatments applied to both, leading [9] and trailing edges [10]. The recent analytical work of Kisil and Ayton [11] proved possible noise reduction through finite porous extensions with a focus on low-frequency gusts and the permeable-impermeable junction. In comparison to other trailing edge modifications, Moreau et al. [12] concluded from their experiments that porous materials offer the best noise control solution. They used a liner-type porous material [13] as an alternative to more complex geometries such as porous metal foams. Geyer et al. [14] experimentally found that fully permeable SD7003 airfoils with different resistivity reduced far-field noise levels by up to 8 dB with respect to solid airfoils. Porous trailing edge treatments fitted to a DLR F16 airfoil were tested by Herr et al. [15] who reported broadband noise reduction up to 2–6 dB depending on the material topology. They proposed air flow resistivity as a key parameter to control the low-frequency noise attenuation. More recently, Rubio Carpio et al. [16] used permeable metallic foams to replace the rigid trailing edge of a NACA0018 airfoil which led to far-field noise reduction of up to 11 dB. Comparison with porous but non-permeable inserts showed that no noise reduction is achieved when flow communication is neglected. This justifies the use of the material's airflow resistivity as a characterizing parameter for potential noise reduction.

Replacing parts of a solid airfoil with permeable materials has been proved effective in reducing airfoil self-noise; however, it introduces an aerodynamic penalty due to increased surface roughness and pressure balance between suction and pressure side. In experiments with poro-serrated trailing edge configurations, Vathylakis et al. [17] reported decreasing lift with respect to the solid reference case while no clear trend towards increasing drag was shown. For partly porous SD7003 airfoils operating at angles of attack between $-12°$ and $24°$, Geyer and Sarradj [18] recently performed balance measurements of lift and drag in addition to acoustic measurements. In agreement with Aldheeb et al. [19], they identified the flow resistivity as characterizing parameter for aerodynamic performance and observed a shift of the drag polar with increasing length of the permeable surface towards higher drag and reduced lift. The present report investigates the possibility to actively control the resistivity of permeable materials by heating of a porous metal foam. The goal is to switch between trailing edge configurations which are in favor of maximum noise reduction and minimum aerodynamic penalty, respectively.

The content of this paper is structured as follows: Section 2 is divided in two parts and first addresses the methods and experimental setup used to establish a relation between fluid temperature and air flow resistivity. The second part issues the aeroacoustic microphone array measurements with a focus on the trailing edge heating capabilities and the acoustic post-processing technique. Experimental results are presented and discussed in Section 3 and a concluding summary of the main findings can be found in Section 4.

## 2. Materials and Methods

### 2.1. Flow Resistivity

Characterization of porous materials can be performed based on properties such as the nominal pore size $d_{p,0}$ of the microscopic structures or porosity $\varphi$. The latter is defined as the ratio of the fluid volume $V_f$ inside the pores divided by the total volume $V_t$ of the porous material and only depends on geometrical properties. In this study, a permeable material consisting of an open-cell, porous metal foam with homogeneously distributed pores is used. The NiCrAl metal foam with a nominal pore size of 800 μm is manufactured by the company Alantum. Figure 1 shows a microscopic image of the foam which is referred to as P800 in the following.

A macroscopic relation between the constant seepage velocity $v_s$ through the material and the related pressure drop $\Delta p$ along the thickness $t$ of the material exists in the form of the Hazen-Dupuit-Darcy relation [20]

$$\frac{\Delta p}{t} = \frac{\mu}{K}v_s + C\rho v_s^2 \,. \tag{1}$$

The seepage velocity $v_s = \dot{Q}/A$ is calculated from the volumetric flow rate $\dot{Q}$ passing through a sample with cross section $A$. The viscous term depends on the dynamic viscosity $\mu$ and the flow permeability $K$. The additional quadratic term represents effects of fluid expansion/contraction and changes in flow direction within porous media. It is referred to as the inertia term and depends on the fluid density $\rho$ and the form coefficient $C$. For low seepage velocities, the quadratic term can be neglected and the ratio $\mu/K$ can directly be related to the pressure drop. A limit for dominant linear behavior can be defined based on the ratio between form forces and viscous forces [20]. The requirement $F_f/F_{visc} < 0.1$ is used here which translates to seepage velocities $v_s < 0.25\,\mathrm{m\,s^{-1}}$ for the given porous material. At $0°$ incidence, the expected pressure difference between suction and pressure side solely depends on the magnitude of turbulent boundary layer pressure fluctuations $p_{rms} \approx 0.1q$ with $q$ being the dynamic inflow pressure [21]. From Equation (1) it follows that the seepage velocity which can be expected for the permeable materials used in this study is low enough to neglect the inertia term. In a previous campaign [16], the physical properties of the metal foam were specified as given in Table 1.

**Table 1.** Characterizing properties of the porous metal foam P800 for a sample thickness of 6 cm as characterized by Rubio Carpio et al. [16].

| Nominal Pore Size $d_{p,0}$ | Porosity $\varphi$ | Resistivity $R$ | Permeability $K$ | form Coefficient $C$ |
|---|---|---|---|---|
| 800 μm | 91.65 % | 6728 Nsm$^{-4}$ | $27.1 \times 10^{-10}$ m$^2$ | 2612.54 m$^{-1}$ |

From the definition of flow resistivity $R = \mu/K$ in the viscous flow regime, it follows that pressure communication can actively be influenced by changing either the material properties (represented by the permeability) or the fluid conditions. The approximate relation between viscosity or air flow resistivity and air temperature can be expressed in the form of the power law

$$\frac{\mu}{\mu_0} = \frac{R}{R_0} = \left(\frac{T}{T_0}\right)^n , \tag{2}$$

where the exponent $n$ depends on the fluid and reference temperature. For air, $n = 2/3$ is commonly chosen [22]. Combining this relation with Eq. 1, it is apparent that the noise reduction capability of permeable materials is a function of fluid temperature. When non-linear effects are taken into account, the pressure drop $\Delta p$ is additionally affected by the fluid density (Equation (1)) which can be related to the temperature via the ideal gas law, according to which $\rho \propto 1/T$.

An experimental characterization of the porous metal foam properties for varying temperatures is performed by using the setup depicted in Figure 1. Pressurized air passes through an electrical heating unit consisting of a coiled heating wire. A desired air temperature up to 90 °C can be controlled manually via adjusting the electric current supply of the heating system.

The heated air is forced through a porous metal foam sample and the static pressure drop along the sample is recorded using a Mensor pressure gauge 2101 in differential mode with a range between $-1000$ Pa and 3500 Pa. The accuracy is $\pm0.01$ % of the full scale (FS) corresponding to 0.45 Pa [23]. The two pressure measurement points are located 5 cm upstream and downstream of the porous sample, respectively. The porous samples have a diameter of 5.5 cm and their thickness is varied from 1 to 6 cm to assess entrance and exit effects on the permeability and form coefficient. In the following, normalized pressure drops from converged measurements with $t = 6$ cm are presented [24].

A Bronkhorst F-202AV mass flow controller (range: 0 to 260 ln/min air; uncertainty: $\pm0.5$ % of measurement value $\pm0.1$ % FS [25]) is used to ensure constant air supply at a prescribed mass flow rate $\dot{m}$. The full-scale contribution to the inaccuracy amounts to $\pm5.6 \times 10^{-6}$ m$^3$ s$^{-1}$ for the given air properties. The air temperature, required to calculate the volumetric flow rate $\dot{Q}$, is measured 1 cm downstream of the porous sample by using a type-K thermocouple. The seepage velocity through the porous sample follows from dividing the volumetric flow rate by the pipe cross-section $v_s = \dot{Q}/A_p$. Assuming ideal gas behavior, the seepage velocity can be expressed in terms of the mass flow rate through

$$v_s = \frac{\dot{m} R_{air} T_\infty}{A_p p_\infty} . \tag{3}$$

For the air temperatures 22 °C, 50 °C and 90 °C, the mass flow rate set-point is varied in a range between $1.5 \times 10^{-4}$ kg s$^{-1}$ and $4.5 \times 10^{-3}$ kg s$^{-1}$ while the pressure drop is recorded. Quadratic least-square curve fitting of the experimental data to Equation (1) in the form $\Delta p = C_a v_s + C_b v_s^2$ is applied to the measurement points in order to retrieve the linear and quadratic coefficients $C_a$ and $C_b$ which can be related to the material parameters in Equation (1).

For every measurement point, the quantities $\dot{m}$ and $\Delta p$ are obtained by averaging 50 data points which are recorded within 10 s. Data points are only acquired for converged flow conditions i.e., when the mass flow rate is within $\pm0.04$ % FS around the set point. From the normally distributed data points, the standard deviation is used to specify the random measurement uncertainty of measured quantities. The standard deviation of temperature, seepage velocity and pressure drop, are obtained by taking error propagation from directly measured quantities into account. The uncertainty of the

curve fitting parameters $C_a$ and $C_b$ can be computed via Monte Carlo sampling with $\Delta p$ and $v_s$ being independent, normally distributed input variables. A total of 1000 simulations is launched and the standard deviation of the coefficients is defined as the interval containing 68 % of the simulated data points. The uncertainty of form coefficient and permeability includes curve-fitting error and random errors from the measurement of the viscosity and density. To simplify the error analysis, it is assumed that the fluid properties and the curve-fitting coefficients are independent variables.

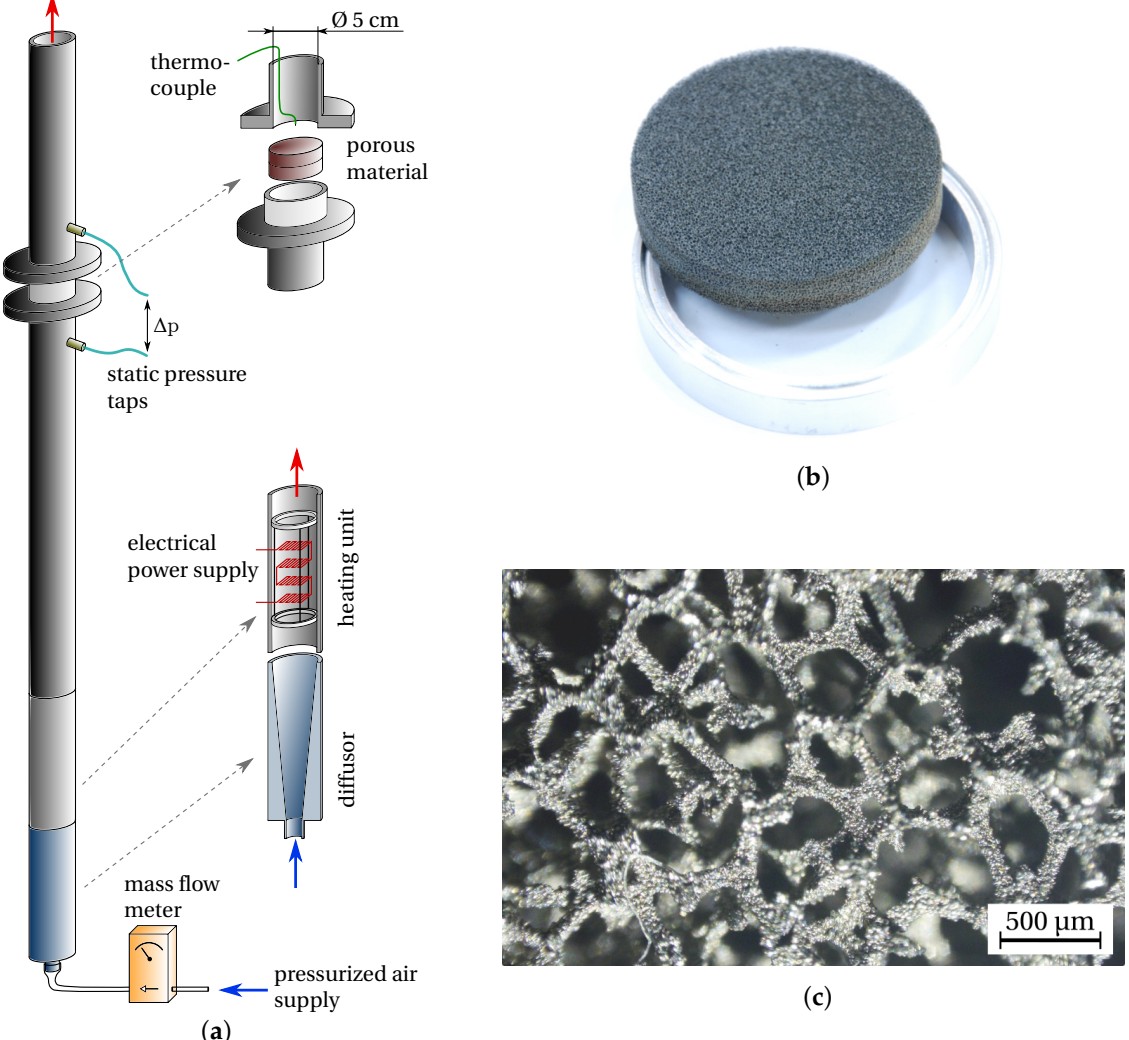

**Figure 1.** Experimental test rig used for specifying porous material properties at varying temperatures (**a**). Pressurized air is forced through a cylindrical material sample (**b**) and the static pressure drop is recorded. The porous material is metallic foam with a nominal pore diameter of $d_p = 800\,\mu\text{m}$ (**c**).

## 2.2. Aeroacoustic Measurements

Acoustic measurements are carried out in the anechoic wind tunnel of Delft University of Technology. The rectangular contraction outlet of the vertical, open-jet wind tunnel measures $40 \times 70\,\text{cm}^2$. TBL-TE noise is evaluated for a NACA0018 airfoil with chord length $c = 20\,\text{cm}$. An interchangeable trailing edge with a length of 4 cm allows for the use of different solid and porous inserts. The airfoil is placed between two wooden side plates with a height of 1.2 m and a distance between the contraction outlet and the leading edge of 0.5 m. All experiments are carried out for a geometrical angle of attack of 0° adjusted based on a digital inclination measuring device.

Experiments are performed for flow speeds between $U = 15\,\text{m s}^{-1}$ and $U = 25\,\text{m s}^{-1}$. For the given airfoil setup and flow conditions, these flow speeds correspond to chord-based Reynolds numbers of $Re_1 = 1.88 \times 10^5$ and $Re_2 = 3.14 \times 10^5$, respectively. To ensure turbulent flow past the

trailing edge, transition is forced at $x/c = 0.2$ using carborundum grains with a diameter of 0.84 mm. Turbulent state of the boundary layer is verified by placing an external microphone at different span- and streamwise locations close to the airfoil surface.

A solid trailing edge insert as well as a porous one made from P800 metal foam is used during the experiments. The porous inserts are manufactured through spark erosion (EDM) machining. Each porous trailing edge is composed out of three individual pieces with spanwise dimensions 13 cm (2x) and 14 cm (1x). Figure 2 shows the geometry of the trailing edge inserts and gives an overview over the different porous versions that are tested.

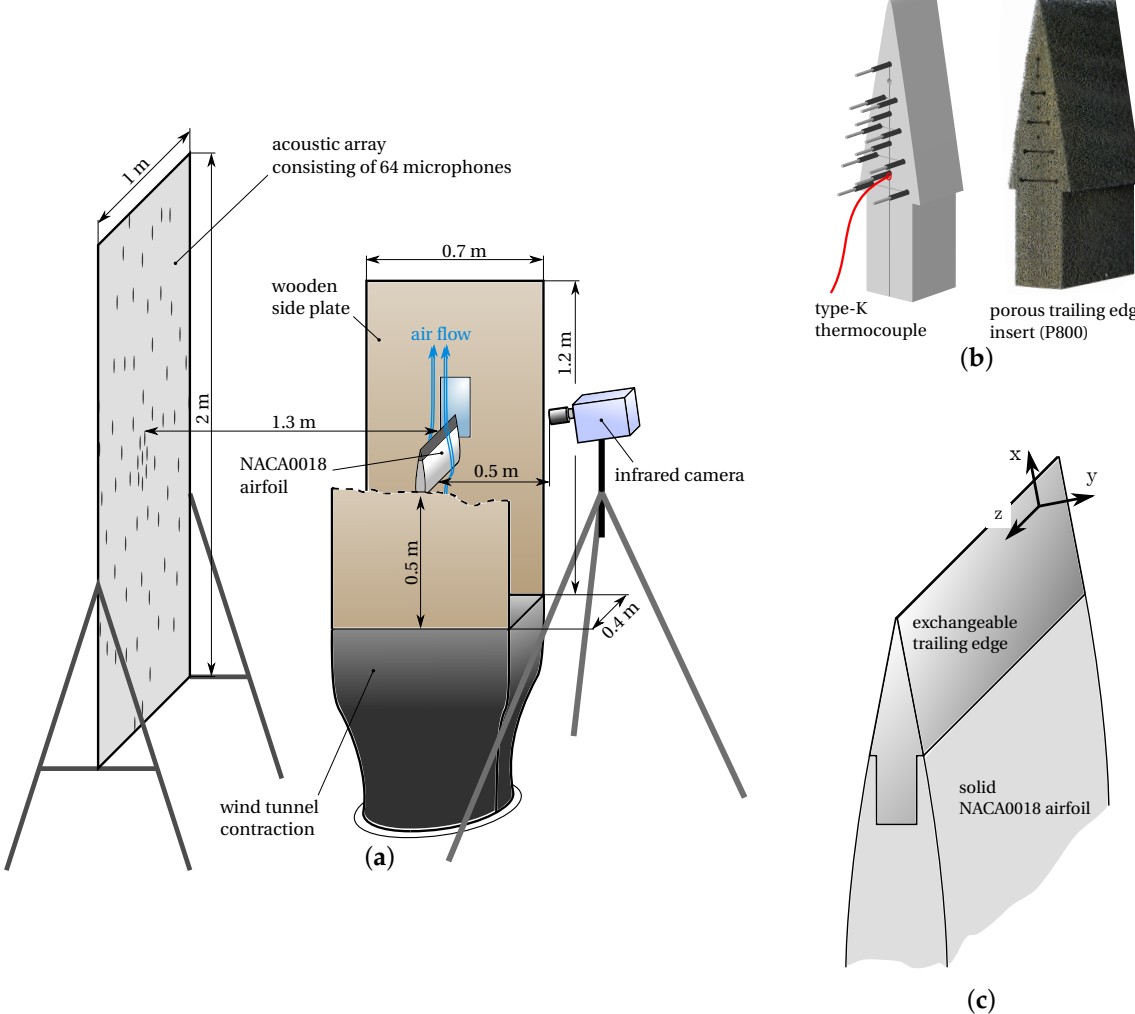

**Figure 2.** Aeroacoustic measurement setup in the vertical, anechoic wind tunnel of TU Delft (**a**). The airfoil can be equipped with exchangeable trailing edge inserts (**c**) and heating of the porous metal foam is achieved by means of electrical heating wires (**b**).

For activation of the porous material via temperature control, the porous trailing edge inserts are equipped with electrical heating wires. A total of 12 heating wires, with an outer diameter of 1.1 mm each, are aligned in the spanwise direction and distributed across the trailing edge cross-section according to Figure 2. Each individual heating wire consists of a conductive inner wire with a diameter of 0.6 mm, covered with an electrical insulation. The core is manufactured out of a copper-nickel alloy (CuNi44) with a high length-specific electrical resistance of $R_s = 1.73\,\Omega\,\text{m}^{-1}$. For insulation from the metallic foam, a polymeric (PVDF), high-temperature shrinking tube is used.

The dissipated heating power $P_{diss} = RI^2$ depends on the wire resistance $R$ and is a quadratic function of the electric current $I$. The resistance of a single wire spanning the entire trailing edge

width of 0.4 m amounts to $R_w = 0.4R_s = 0.69\,\Omega$ and with a maximum current of $I = 5\,\text{A}$, the available heating power results in $P_{el} \approx 210\,\text{W}$.

The surface temperature of the trailing edge is analyzed using a CEDIP Titanium 530L IR infrared camera with a resolution of $320 \times 256$ pixels. The camera lens is positioned at a distance of 0.5 m away from the airfoil plane. The heating wire temperature is measured using a type-K thermocouple placed inside the porous material close to the wire surface as shown in Figure 2. The measured surface temperature distribution is plotted in Figure 3 for three different flow speeds. The permeable trailing edge material promotes strong forced convection resulting in effective surface cooling at higher flow speeds. Especially towards the trailing edge tip, the surface temperature of the material approaches ambient conditions. Furthermore, the poor heat conduction, resulting from the high porosity of the material, and the lack of heating wires in the trailing edge tip region prevents homogeneous heating. For the maximum heating power input of $P_{el} = 210\,\text{W}$, the wire surface temperature reaches approximately $270\,°\text{C}$ which is much higher than the surface temperature.

For quantification of the emitted TBL-TE noise from different trailing edge configurations, a phased microphone array is used to localize the trailing edge noise source and isolate it from interfering background noise sources. The acoustic array consists of 64 G.R.A.S. 40PH free-field microphones with built-in preamplifiers. The microphones are distributed according to a variation of Underbrink's design [26] consisting of seven logarithmic spiral arms and one microphone in the center. The array is located 1.3 m away from the airfoil plane. In order to optimize the array resolution in vertical direction, the Underbrink design is stretched. The microphones are spread across a 1 m wide and 2 m high plane and the array layout is illustrated in Figure 4.

For each test configuration, the microphone recording time amounts to 60 s at a sampling frequency of 50 kHz. The raw signals are split into time windows of $\Delta t = 0.1\,\text{s}$ which results in a frequency resolution of $\Delta f = 10\,\text{Hz}$. The data in every time window are weighted using a Hann function and a window overlap of 50 % is chosen.

A Fourier transform is applied to the time domain sound pressure levels of every microphone in order to determine the complex pressure amplitudes $p(f)$ as a function of frequency. For each of the 1199 windows, the spectral densities of the 64 microphones are linked by calculating the Cross Spectral Matrix $CSM_k = 0.5p(f_k)p(f_k)^*$ for a certain frequency $f_k$ of interest. Results from the individual windows are averaged in order to obtain a single CSM for every frequency. For the given experimental setup, the range $f_{\min} = 500\,\text{Hz}$ up to $f_{\max} = 5000\,\text{Hz}$ is chosen during the post-processing.

A conventional delay-and-sum beamforming method in the frequency domain [27] is employed to localize the noise source and its strength in the airfoil aligned scan plane of interest. The dimensions of the scan plane amount to 2 m in height and 1 m in width with a resolution of 0.01 m. The frequency-dependent resolution of the acoustic camera follows from Rayleigh's limit which can be written as [28]

$$\Delta x = l \tan\left(1.22 \frac{c_0}{fD}\right) , \tag{4}$$

where $\Delta x$ is the smallest resolvable length scale, $l$ is the distance between the array and the scan plane, $c_0$ is the speed of sound and $D$ is the diameter of the circular array. For the highest frequency of interest $f_{\max} = 5000\,\text{Hz}$ and under the assumption of $D = 2\,\text{m}$ in the vertical direction, a minimum value of $\Delta x_{min} = 0.05\,\text{m}$ results which justifies the chosen scan grid resolution.

For every scan grid point $j$, a frequency-specific steering vector is constructed which represents the received signal of microphone $n$ emitted from a modeled monopole source (ideal point source) located at point $j$ in the scan plane. The steering vector incorporates the distance between emitter and receiver as well as the resulting time delay of the signal. The corresponding amplitudes of the steering vectors are obtained in the form of the source autopowers by minimizing the difference between the recorded and modeled pressure data.

In quiescent air, the signal time delay directly follows from the speed of sound $c_0$ and the distance between scan point and receiver. For measurements in open-section wind tunnels, where the

microphone array is located outside the flow, the emitted sound waves pass regions with different air speeds as well as the non-uniform shear layer. As suggested by Sijtsma [29], uniform flow with an effective Mach number of

$$M_{x,eff} = M_x \frac{y_{sl}}{y} .$$ (5)

can be assumed. The relation is given for noise propagation in the *y*-direction perpendicular to the vertical component of the Mach number $M_x$. For the distance $y = 1.3$ m between scan plane and array, a shear layer distance of $y_{sl} = 0.5$ proved to be appropriate. For the given experimental setup, this value results in a flow speed independent position of the main noise source.

The beamforming results are further used to obtain the spectrum of TBL-TE noise emitted by the airfoil. Source Power Integration (SPI) [30] is applied with the intention to isolate TBL-TE noise from background sources such as noise from interactions between the flow and the test section side plates. For SPI, an integration area in the scan plane is defined according to Figure 4. Centered on the trailing edge, the integration area extends 0.13 m in streamwise direction and 0.2 m in spanwise direction. The integration area should include the complete main lobe of TBL-TE noise for the whole frequency range of interest while excluding side lobes and contributions from other noise sources.

The experimentally obtained source autopowers are summed up within the integration area and an array-specific scaling factor is applied as suggested in [30]. From the corrected, integrated source powers, the averaged sound pressure levels (SPLs) over 1/3-octave bands at a distance of 1 m from the scan plane are calculated. For visualization of the beamforming results, the ratio between source autopowers $A_j$ and a reference pressure of $p_0 = 2 \times 10^{-5}$ Pa are converted into SPLs as received at a distance of $r = 1$ m from the scan plane

$$\mathrm{SPL}_j = 20 \log_{10} \frac{\sqrt{A_j}}{r p_0} .$$ (6)

Aeroacoustic measurements are repeated five times for every test case and plotted band levels represent the average of these runs. Repeatability of the results is proven by alternately recording noise spectra of heated and unheated cases.

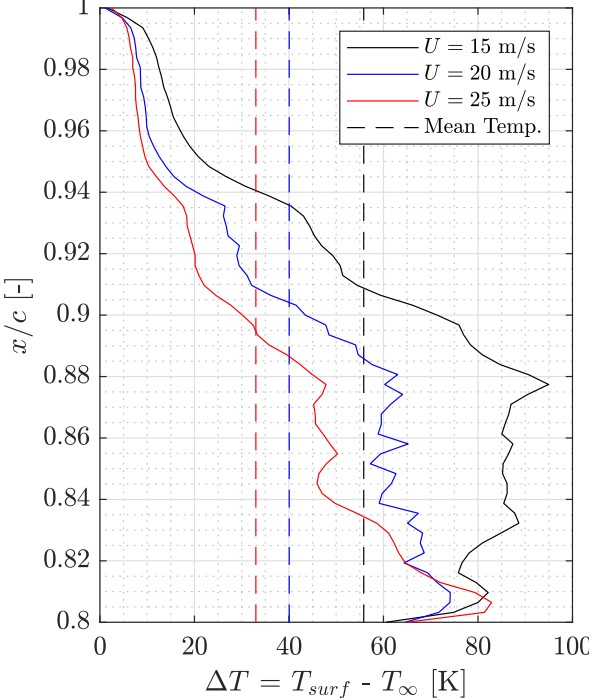

**Figure 3.** Temperature distribution of heated porous trailing edge for different flow speeds.

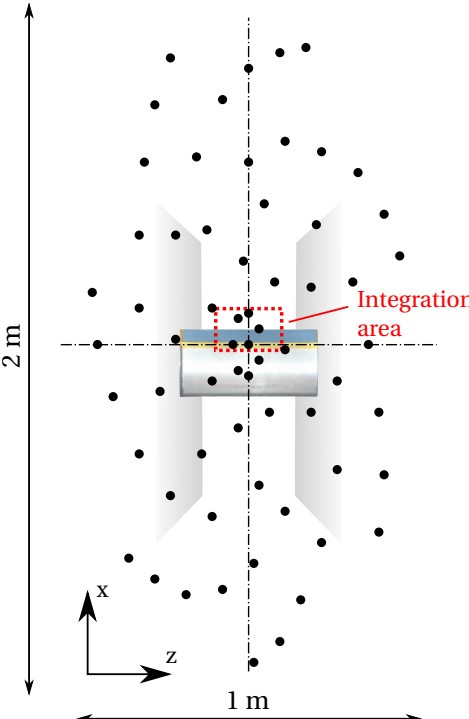

**Figure 4.** Microphone array setup and location of integration area.

## 3. Results and Discussion

### 3.1. Heated Flow through Porous Materials

The influence of fluid temperature on the normalized pressure drop of a sample stack with $t = 6$ cm is shown in Figure 5 for the temperatures $T = 22, 50$ and $90\,°$C. A magnification of the linear flow region $v_s < 0.25\,\mathrm{m\,s^{-1}}$ shows that the normalized pressure drop increases with fluid temperature while the opposite behavior can be observed for higher seepage velocities. Results agree with the predictions based on Equation (1), where the viscosity and density affect the linear and quadratic term, respectively. The variation of form coefficient $C$, permeability $K$ and flow resistivity $R$ with temperature is given in Table 2. Figure 6 illustrates the relation between non-dimensional coefficient ratios and temperature ratios. Error bars indicate the standard deviation of the simulated temperature and coefficient ratios. The power-law based relation between resistivity and temperature (Equation (2)) is plotted for $n = 0.6$. The experimental results follow the same trend but indicate a slight offset from the expected relation. At the higher temperature ($T = 90\,°$C), measurements show better agreement with the theoretical prediction.

**Table 2.** Characterizing permeable (P800) material coefficients as a function of fluid temperature.

|  | $T = 22\,°$C | $T = 50\,°$C | $T = 90\,°$C |
|---|---|---|---|
| $R = C_a\ [\mathrm{Ns/m^4}]$ | 6274 | 6511 | 7003 |
| $K = \mu/R\ [\mathrm{m^2}]$ | $2.92 \times 10^{-9}$ | $3.00 \times 10^{-9}$ | $3.05 \times 10^{-9}$ |
| $C = C_b/\rho\ [\mathrm{m^{-1}}]$ | 2242 | 2202 | 2124 |

The effects of fluid temperature on permeability and form coefficient are determined by correcting the fitting coefficients $C_a$ and $C_b$ for the change in fluid viscosity and density, respectively. From the experimental results it can be inferred that the fluid independent coefficients $C$ and $K$ are not independent from temperature. At $T = 90\,°$C the deviation with respect to ambient conditions lies in the order of 5 % for both coefficients. The variation with temperature is attributed to systematic errors which are likely caused by thermal expansion of the measurement setup. Thus, a clear temperature

dependency of the geometrical constants cannot be derived even though the expected values $K/K_0 = 1$ and $C/C_0 = 1$ do not lie within the calculated, random uncertainty limits. From the uncertainty analysis it can be concluded that the random measurement errors affect the linear coefficient $C_a$ less than the quadratic coefficient $C_b$ which is reflected in the error bar lengths in Figure 6.

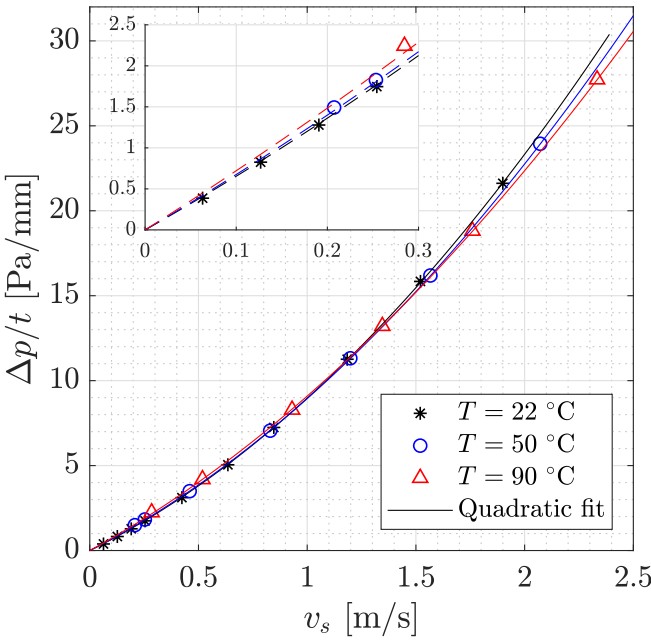

**Figure 5.** Change in normalized pressure drop along porous samples with $t = 6\,\text{cm}$ due to changing fluid temperatures. Quadratic curve fitting is indicated by the dashed lines and a magnification of the linear flow region is provided.

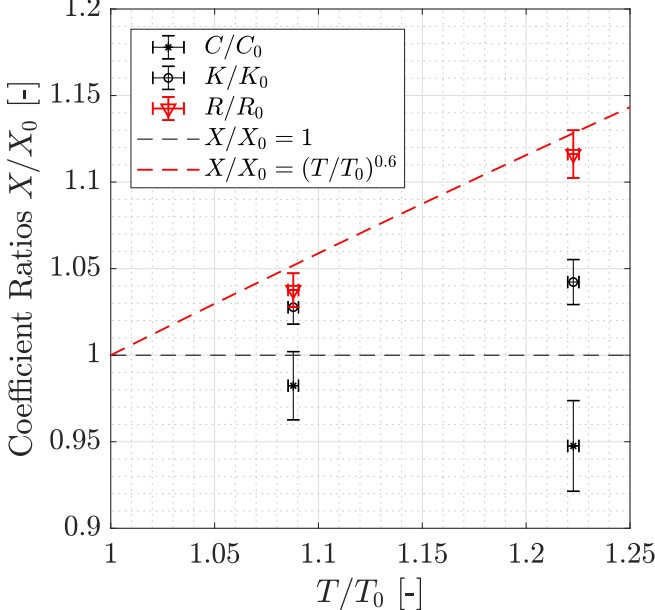

**Figure 6.** Change of material constants $C/C_0$, $K/K_0$ and $R/R_0$ with temperature for $t = 6\,\text{cm}$. Black dashed line indicates the constant coefficient ratio as expected for the geometrical constants $C$ and $K$. The theoretical relation for $R$ according to Equation (2) with $n = 0.6$ is indicated in red. Error bars represent regions which contain 68 % of simulated values.

### 3.2. Far-Field Noise

A comparison of far-field noise spectra from porous and solid trailing edges is displayed in Figure 7 to demonstrate the frequency region in which noise mitigation occurs. The SPLs are calculated according to Equation (6) for two different flow speeds and 1/3-octave band levels are plotted against the frequency. In agreement with previous campaigns [14–16,31], low-frequency noise is effectively reduced by up to 10 dB within the lowest evaluated frequency band centered around 630 Hz. Above a frequency of 1.3 kHz for $U = 15\,\mathrm{m\,s^{-1}}$ and 2 kHz for $U = 25\,\mathrm{m\,s^{-1}}$, the use of a porous trailing edge leads to noise increase. The occurrence of excess noise is commonly attributed to the higher surface roughness of porous materials [14,15]. Above a frequency of 1.25 kHz, the slope of the porous trailing edge noise spectra changes with respect to the linear trend at lower frequencies.

The deviation between far-field spectra from homogeneous, porous inserts and porous inserts equipped with heating wires is depicted in Figure 7 to show wire installation effects. In addition to absolute SPL levels, the difference between the homogeneous baseline and the porous inserts fitted with wires is plotted in the form of

$$\Delta SPL_{1/3} = SPL_{Wire} - SPL_{Hom.} . \tag{7}$$

The installation of heating wires inside the P800 metal foam affects the porosity $\varphi$ of the material since a certain volume of air is replaced by the non-permeable wires. Far-field noise levels increase by approximately 1 dB in the low-frequency range upon heating wire installation while high-frequency bands are less affected. Based on an aeroacoustic characterization of porous materials, Herr et al. [15] reported high-frequency excess noise which occurred when large rigid structures were embedded in porous materials and the use of solid wires in this study is expected to reduce the communication between suction and pressure side, hence deteriorating noise abatement performance.

Results from the acoustic measurements confirm that active change of TBL-TE noise characteristics can be achieved by changing the temperature of the porous trailing edge. Far-field noise spectra for two different flow speeds are included in Figure 7. A broadband increase in noise levels is observed for increasing trailing edge temperature except for the case of higher flow velocity in the range between $f_c = 2\,\mathrm{kHz}$ and $f_c = 2.5\,\mathrm{kHz}$. Effects from changing temperatures are more pronounced at $U = 15\,\mathrm{m\,s^{-1}}$ where a maximum noise increase of almost 2.5 dB can be observed. Given the maximum heating input of 210 W, measurements at higher flow speeds are stronger affected by cooling effects due to forced convection and achievable far-field noise level changes are lower. The maximum activation effect is observed in the frequency band centered around $f_c = 1\,\mathrm{kHz}$ and $f_c = 1.25\,\mathrm{kHz}$ for $U = 25\,\mathrm{m\,s^{-1}}$ and $U = 15\,\mathrm{m\,s^{-1}}$, respectively.

The dominant activation mechanism which leads to the variation in far-field noise levels is assumed to be the resistivity change of the trailing edge material upon heating. Cross-flow through the permeable trailing edge is required if communication between the suction and pressure side occurs. Assuming that this seepage flow is effectively heated while penetrating the porous material, pressure communication is influenced. The decrease in noise abatement performance upon activation is in agreement with the increase in resistivity with temperature reported in Section 3.1.

The far-field noise spectra of heated trailing edges indicate that the frequency bands around 2 kHz and 2.5 kHz are less affected by resistivity changes. This frequency region coincides with the rising noise levels in the spectra of unheated porous trailing edges which are attributed to surface roughness noise. Interestingly, the same trend of broadband noise increase and smoothing of the spectral curve is present in the analysis of heating wire installation effects. Due to a temperature gradient of roughly 150 °C between the heating wires and the trailing edge surface, the change in material resistivity is not uniform but mostly affects the region close to the heat sources.

In a number of experimental studies, the effect of changing permeability on far-field noise spectra was evaluated [14,15,32,33]. For a resistivity increase of less than 50 %, only marginal changes in emitted noise levels (in the order of 1 dB) can be expected according to these studies. However,

transferring their observations directly to the present results from trailing edge heating is flawed, firstly because resistivity changes were simulated by modifying the geometrical structure of the foam which remains unaffected here. Thermal expansion of the material, and therefore active change of the geometrical pore size, is negligible in this study which can be deduced from microscopic images of heated porous samples. Secondly, the temperature gradients inside the metal foam result in a non-uniform distribution of flow resistivity. Critical analysis of the results indicates that additional effects other than resistivity variations could play a role. Especially for the heated inserts at $U = 15\,\mathrm{m\,s^{-1}}$, broadband noise levels *exceed* the solid trailing edge noise even in the mid-frequency regime. Even though a lack of communication between suction and pressure side would affect the mitigation mechanism negatively, it cannot account for increasing noise beyond the reference case. It is expected that the non-uniform trailing edge temperature aggravates heating wire installation effects due to the occurrence of non-permeable regions within the inserts.

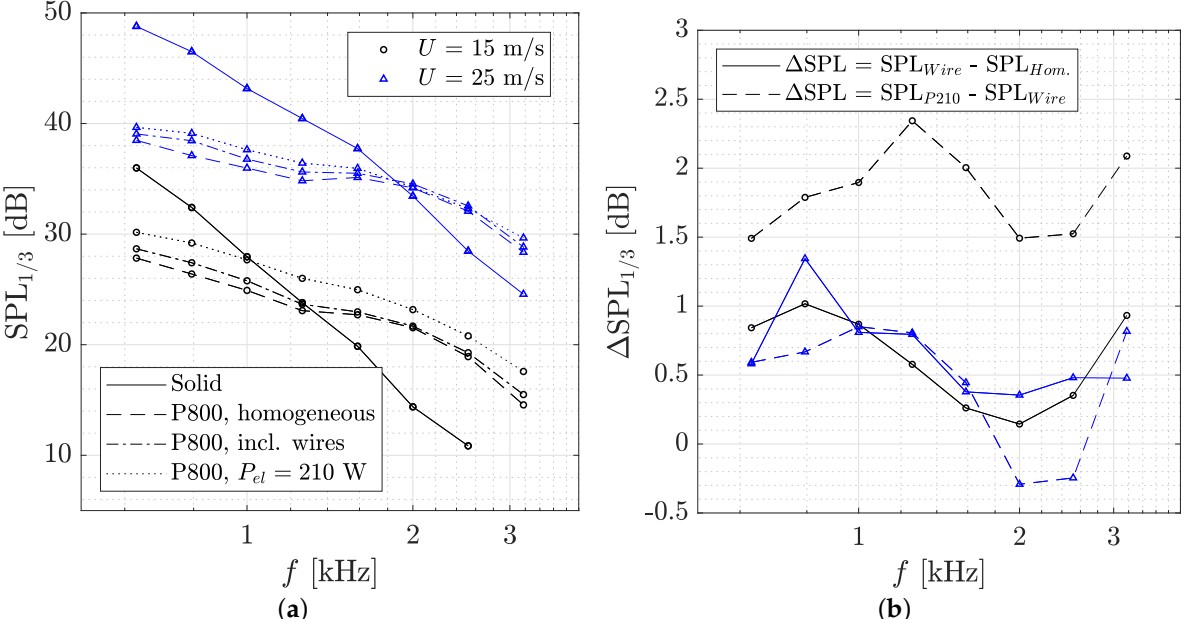

**Figure 7.** Comparison of far-field noise spectra of different heated and unheated trailing edge cases for $U = 15\,\mathrm{m\,s^{-1}}$ and $U = 25\,\mathrm{m\,s^{-1}}$. SPLs as observed at a distance of $1\,\mathrm{m}$ from the airfoil are depicted (**a**) as well as a comparison between selected measurement cases (**b**).

## 4. Conclusions

The static pressure drop that is experienced by constant flow through permeable materials is a function of geometrical and fluid properties. In this work, the focus lies on actively changing the flow resistivity of porous metal foams by means of flow heating. Based on a simple power-law relation between fluid temperature and viscosity, the material resistivity increases with fluid temperature according to $R/R_0 \approx (T/T_0)^{0.6}$. Using a resistivity rig with built-in heating unit, this relation is observed in good approximation.

The emitted TBL-TE noise levels of a NACA0018 airfoil with different trailing edge inserts are measured in the anechoic wind tunnel at TU Delft. Equipping the porous trailing edge with heating wires results in increased broadband noise levels of up to $1.5\,\mathrm{dB}$. Upon heating, active change of noise mitigation characteristics is observed. For an average trailing edge surface temperature of $57\,^{\circ}\mathrm{C}$ above ambient conditions and an inflow velocity of $U = 15\,\mathrm{m\,s^{-1}}$, far-field noise band levels increase by up to $2.5\,\mathrm{dB}$. Strong temperature gradients along the trailing edge surface and inside the foam indicate effective heat transfer between the metal foam and the penetrating fluid. Having shown that the flow resistivity increases with the fluid temperature, it is concluded that hampered flow communication across the permeable inserts is a main factor for observed active change of TBL-TE noise emissions.

The increase of broadband noise levels with rising fluid temperature shows similar characteristics compared to the wire installation effects. It is therefore argued that not only a uniform change of material resistivity accounts for the magnitude of noise increase but also a non-uniform resistivity distribution with peak values close to the wires. Further sources of excess noise upon heating, such as the change in density experienced by convected pressure fluctuations over the heated trailing edge, could play an additional role in the observed activation behavior.

Embedding heating wires in porous trailing edges of aircraft wings or wind turbine blades might not be feasible due to the high complexity it adds to the design. Future experiments could focus on uniform trailing edge heating which would provide the ability to actively influence flow resistivity *independently* of the microscopic pore structure. This would allow identifying beneficial pore geometry effects while keeping the material resistivity constant.

**Author Contributions:** Conceptualization, methodology as well as experimental investigation was performed by J.M. and A.R.C. Responsible author was J.M. while original draft preparation was supported by A.R.C. and D.R. Topic proposer and main supervisor of the project was D.R.

**Acknowledgments:** The authors would like to thank the technical staff in the low-speed wind tunnel lab for their support during the experimental campaign. Special thanks to Reza Hedayati from the Novel Aerospace Materials department for his advice in the field of porous materials. The evaluation of beamforming results was a crucial part of this project and we would like to thank Salil Luesutthiviboon from the Aircraft Noise and Climate Effects department for his help. This work was carried out within the IPERMAN project group and many thanks go to the multidisciplinary team members and their technical advice.

**Conflicts of Interest:** The authors declare no conflict of interest. The funders had no role in the design of the study; in the collection, analyses, or interpretation of data; in the writing of the manuscript, or in the decision to publish the results.

## Abbreviations

The following abbreviations are used in this manuscript:

| | |
|---|---|
| CSM | Cross Spectral Matrix |
| EDM | Electrical Discharge Machining |
| FS | Full Scale |
| IR | Infrared |
| P800 | Porous metal foam with nominal pore size of $d_p = 800\,\mu m$ |
| SPI | Source Power Integration |
| SPL | Sound Pressure Level |
| TBL-TE noise | Turbulent Boundary Layer Trailing Edge noise |

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
