# Peer review of "Temperature-Activated Change of Permeable Material Properties for Low-Noise Trailing Edge Applications"

_applsci, doi:10.3390/app9153119_

Round 1
Reviewer 1 Report
This study has investigated the properties changes of permeable material upon heating at different temperature. The broadband noise emission can be improved by increase the temperature of the porous trailing edge inserted. The paper can be published in applied sciences Journal but need some improvement. Below some questions and concerns listed for the author:
Q1. In the experiment, have you consider the air environment of the noise scattering? Does the experiment controlled in different gas phase environment?
Q2: Does the noise scattering and the temperature effect only happened on the edge?
Q3: Have author consider the properties of the metal foam used in the study? does it related with the property of the material itself, such as thermo expansion?
Q4: From the data, the resistant of the permeable (P800) material improved from 22 to 90C.Is there a reason for this temperature range?
Q5: What is the mechanism of the noise scattering properties change upon temperature? Can author tried to explain.
Q6: How to apply this phenomenon in use this material in the noise control or other applications?
Reviewer 2 Report
Firstly, my warmest congrats to the Authors for the work done. The manuscript is written very well, the quality of the figures is extremely high as well as their clarity. The reading flows is good and I've enjoyed reading it.
I have only a few minor points:
- Abstract: lines 8-10: "Differences in noise...". I suggest to remove this sentence and mention it only in the Conclusion section.
- Materials and Methods: "the accompanied pressure drop", better "the related pressure drops"?. Please also add a reference for "Hazen-Dupuit-Darcy" relation.
- Line 206, it can be due also to additive noise/sensitivity changes of the thermal camera, especially if it is an uncooled model.
- Please, add more recent references throughout the manuscript.
Queries:
- line 144: 270 °C maximum. Can this temperature provoke local damage to the material structure itself?
- It can be good to mention if the proposed strategy, i.e. embedding wires into the structure, would be feasible in real in-life service application.
Round 2
Reviewer 1 Report
The study in this manuscript has mainly work on the analysis the property variation in noise scattering. Although it doesn't have a lot of novelty, it may provide a parameter for other related study. I suggest it can be published as current form.